# Altering the Sex Pheromone Cyclo(l-Pro-l-Pro) of the Diatom *Seminavis robusta* towards a Chemical Probe

**DOI:** 10.3390/ijms22031037

**Published:** 2021-01-21

**Authors:** Eli Bonneure, Amber De Baets, Sam De Decker, Koen Van den Berge, Lieven Clement, Wim Vyverman, Sven Mangelinckx

**Affiliations:** 1Department of Green Chemistry and Technology—SynBioC, Faculty of Bioscience Engineering, Ghent University, Coupure Links 653, 9000 Ghent, Belgium; eli.bonneure@ugent.be (E.B.); amber.debaets@gmail.com (A.D.B.); 2Department of Biology—Protistology and Aquatic Ecology, Faculty of Sciences, Ghent University, Krijgslaan 281/S8, 9000 Ghent, Belgium; samddecker@gmail.com (S.D.D.); wim.vyverman@ugent.be (W.V.); 3Department of Applied Mathematics, Computer Science and Statistics, Faculty of Sciences, Ghent University, Krijgslaan 281/S9, 9000 Ghent, Belgium; koen.vandenberge@ugent.be (K.V.d.B.); lieven.clement@ugent.be (L.C.)

**Keywords:** diatom, pheromone, diketopiperazine, chemical probe, diazirine

## Abstract

As a major group of algae, diatoms are responsible for a substantial part of the primary production on the planet. Pennate diatoms have a predominantly benthic lifestyle and are the most species-rich diatom group, with members of the raphid clades being motile and generally having heterothallic sexual reproduction. It was recently shown that the model species *Seminavis robusta* uses multiple sexual cues during mating, including cyclo(l-Pro-l-Pro) as an attraction pheromone. Elaboration of the pheromone-detection system is a key aspect in elucidating pennate diatom life-cycle regulation that could yield novel fundamental insights into diatom speciation. This study reports the synthesis and bio-evaluation of seven novel pheromone analogs containing small structural alterations to the cyclo(l-Pro-l-Pro) pheromone. Toxicity, attraction, and interference assays were applied to assess their potential activity as a pheromone. Most of our analogs show a moderate-to-good bioactivity and low-to-no phytotoxicity. The pheromone activity of azide- and diazirine-containing analogs was unaffected and induced a similar mating behavior as the natural pheromone. These results demonstrate that the introduction of confined structural modifications can be used to develop a chemical probe based on the diazirine- and/or azide-containing analogs to study the pheromone-detection system of *S. robusta*.

## 1. Introduction

Diatoms play a key role in the biosphere. As an important group of phytoplankton, they account for about one-fifth of the photosynthesis on earth and form the basis of many of the aquatic ecosystems that exist on our planet [1,2]. In addition, they present opportunities for biotechnological applications [3,4,5,6].

The benthic diatom *Seminavis robusta* is an experimental model organism for diatom research [7,8]. *S. robusta* has a typical diatom life cycle: the cell size declines during mitotic division and is restored after sexual reproduction, using a heterothallic mating system (Figure 1) [9]. Once the cell size reaches the sexual size threshold, both mating types start producing a sex inducing pheromone (SIP). The presence of SIP^+^ induces mating type – (MT^−^) to produce an attraction pheromone: the diketopiperazine cyclo(l-Pro-l-Pro) (**1**) (Figure 2) [10].

The diketopiperazine scaffold is present in a plethora of metabolites [12,13,14]. Regardless of the broad occurrence of these cyclopeptides, their receptors are not extensively characterized. Bilcke et al. [11] found that in *S. robusta* MT^+^ genes encoding for transmembrane proteins were upregulated in response to SIP^−^, suggesting that a G protein-coupled receptor (GPCR) is responsible for the perception of **1**.

We evaluated the possibility of modifying the pheromone to a chemical probe, in order to identify the putative receptor. However, certain structural restrictions should be taken into account. Lembke et al. [15] described that the diketopiperazine structure of **1** can be modified while retaining its activity, but only if both of the flanking ring structures were present. The authors also tested a hydroxyl-substituted analog of **1** (cyclo(4-OH-Pro-4-OH-Pro) that proved to be inactive. This observation demonstrates the confined chemical space in which the pheromone is positioned.

Photoaffinity labeling has, since it was first reported in 1962 by Westheimer et al. [16], provided the scientific community with a broad array of tools to study ligand-protein interactions [17,18,19] and has proven to be useful for GPCR characterization [20,21,22,23]. In the case of diketopiperazine **1**, the (alkyl) diazirine is an attractive photoreactive group. While having a lower cross-linking efficiency in comparison to other photoreactive groups, diazirines have a small footprint. Therefore, we deemed that the inclusion of an alkyl diazirine would be the best option to render a bioactive probe.

Photoaffinity labeling (PAL) probes are more useful if means are provided to conjugate a reporter group to the cross-linked target receptor. An elegant approach is the use of click chemistry, which has been widely adopted since its introduction by Sharpless et al. in 2001 [24,25,26,27]. The inclusion of a click-chemistry compatible moiety, such as an azide, could allow for an easy detection of the target.

A chemical probe derived from **1** could also be applied in other disciplines. The diketopiperazine was reported to be a constituent of pheromonal secretions in two other species, namely in the femoral gland of the *Sceloporus virgatus* lizard [28] and the mating plug of the *Bombus terrestris* bumblebee [29], and was also identified in the *Aspergillus fumigatus* fungus, the *Lucilia sericata* blowfly and the Antarctic *Pseudoalteromonas haloplanktis* bacterium [28]. Recently, **1** was also shown to be an effective induced resistance stimulus for rice plants [30], triggering a systemically enhanced defense against the root-knot nematode *Meloidogyne graminicola*.

## 2. Results

New pheromone analogs were synthesized according to literature protocols. We integrated chemical moieties that could be used for photo-cross-linking and click chemistry and evaluated the modifiability of the pheromone scaffold at the C-4 position of one of the two proline rings (Figure 2). *Trans*-4-hydroxy-l-proline (**2**, Figure 2) is an inexpensive commercial analog of l-proline and allows a straightforward modification at the C-4 position.

After synthesis and purification, the phytotoxicity of the analogs was examined and comparative studies were performed to assess the equivalence of the analogs and the natural pheromone. The comparative studies were based on an interference assay, an attraction assay and a short evaluation of the conformation of the diketopiperazines.

### 2.1. Diketopiperazine Synthesis

Seven diketopiperazines (**3** to **9**, Figure 3) were synthesized in close resemblance to **1**. These pheromone analogs contain the same cyclo(l-Pro-l-Pro) backbone with an additional functional group attached onto the C-4 of one of the l-proline rings, consisting of a diazirine (**3**), an azide (**4** and **5**), a hydroxyl (**6**), a methoxy (**7**), and an acetal (**8**). In analog (**9**), one of the proline rings was substituted with its four-membered azetidine equivalent.

Prior to diketopiperazine synthesis, a small library of six l-proline analogs was synthesized, starting from commercial *trans*-4-hydroxy-l-proline **2** as shown in Scheme 1A. A Boc group was introduced [31] and oxoproline **11** and diazirinylproline **12** were subsequently synthesized according to Van der Meijden et al. [32]. Azidoprolines **16** and **18** were synthesized using a method adopted from Marusawa et al. [33]. Methyl ester **13** was synthesized using the method of Chalker et al. [34].

Diketopiperazines **1** and **3**–**9**, presented in Figure 3, were synthesized with a method modified from Campbell et al. [35], of which a general overview is given in Scheme 1B. OxymaPure [46,47] was used as additive and DMF was used as solvent during the coupling step. Diketopiperazines **1** and **9** were synthesized by coupling H-Pro-OMe·HCl (Apollo Scientific) with respectively Boc-Pro-OH [31] and Boc-Aze-OH (ChemPur). Proline analogs **11** and **12** were coupled with H-Pro-OMe·HCl to their corresponding diketopiperazines **8** and **3**, with analog **11** undergoing acetal formation during the acidic Boc-removal in methanol. Proline analogs **16** and **18** were coupled with Boc-Pro-OH after being treated with TFA in DCM or HCl in methanol, yielding respectively **5** and **4**. H-Hyp-OMe·HCl (**13**) was coupled with Boc-Pro-OH and methylated using MeI towards **7** [36]. Proline analog **17** was coupled, after Boc-removal with hydrochloric acid, with Boc-Pro-OH, yielding **6**.

### 2.2. Diketopiperazine Phytotoxicity

Diketopiperazines are a large class of compounds with a broad range of activities [12]. Recently, some diketopiperazines were reported to be algicidal [48,49]. For this reason, it was not implausible that small changes to the pheromone scaffold could induce algicidal effects and thereby could affect the physiology of *S. robusta*.

We determined the short-term phytotoxicity of the newly synthesized analogs, as the exposure time to the pheromone analogs during the bioassays is limited. Pulse Amplitude Modulation (PAM) fluorometry was used to measure the chlorophyll fluorescence of photosystem II (PSII), a parameter that is related to the overall fitness of algae [50]. The technique has been successfully exploited to assess toxic effects of herbicides [51,52], heavy metals [53], volatile organic compounds [54], pharmaceuticals [55], and quorum sensing related compounds [56] towards microalgae.

Cultures of *S. robusta* were treated with 10 µM of **1**, **3**–**9** and with 1 µM of DCMU, a PSII inhibitor [57,58]. The quantum yield of PSII (Y_II_) was measured for 30 min and the resulting data of the analogs was interpolated between **1** (set to 0% inhibition) and DCMU (100% inhibition) (Figure 4, see Appendix A for original Y_II_ data).

The inhibition of Y_II_ was significant for methoxy-substituted analog **7**, with an average inhibition of 10%. The other pheromone analogs can be categorized in two groups. The first group comprising of compounds **3**, **5,** and **9** shows little phytotoxicity, with an average inhibition of 6, 3.9, and 4.8% respectively at 10 µM (*p* > 0.13, Figure 4). The second group, consisting of compounds **4**, **6,** and **8**, shows an average inhibition of less than 1%.

### 2.3. Interference Assay

The pheromone potency of the analogs was evaluated according to an interference assay, in which the attraction of MT^+^ towards a point source of the natural pheromone is disturbed by the presence of a pheromone analog [10,15]. The MT^+^ cells were conditioned with spent medium from a MT^−^ culture, that was evaluated for the presence of the SIP^−^ conditioning factor [10].

Diketopiperazines **3** to **9** were tested in concentrations of 10 nM, 100 nM, and 10 µM in three independent experiments. Treatments with compound **1** were included in every experiment, allowing us to compare the outcome of the pheromone analogs interexperimentally. Oasis HLB beads, that were coated with a load of 2 nmol of **1** per mg of beads, were used as a pheromone point source.

Each experiment included two control treatments. In the first control treatment (positive control), no pheromone (analog) was added. The attraction towards the coated beads was not disturbed and maximal. For a second control treatment, blank beads were added (negative control) to measure coincidental contact of the algae with the beads. The total amount of treatments per experiment was 11 or 14, equaling to respectively three or four analogs at three concentrations and two control treatments. In the case of experiments A and B, two replicates were provided per multi-well plate, occupying 22 of the available 24 wells. In experiment C, every multi-well plate was provided with only one replicate (14 out of 24 wells). The activity screening of the pheromone analogs was spread over three independent experiments according to their availability.

The interference caused by adding **1** to the culture medium (Figure 5) was significant for all three of the tested concentrations, except in experiment C. A normal dose-response relationship, where a low concentration yields a limited interference and a high concentration yields an increased interference, was expected. However, it was noted that the dose-response trend of **1** was variable.

The pheromone analogs were also compared to **1** at 100 nM, the physiological concentration of the pheromone (Figure 6). The analogs’ estimated average fraction of attractive beads was interpolated between the positive control (no interference) and **1** at 100 nM (set as 100% interference). The interpolation makes it possible to compare the pheromonal potency of the analogs relative to the natural pheromone and irrespective of the experiment in which they were measured (see Appendix A).

### 2.4. Attraction Assay

Diketopiperazines **3**, **4,** and **5** were also evaluated using an attraction assay, where the attraction towards pheromone-analog-coated beads was assessed (Figure 7). Beads were coated in the same manner as was previously done with diketopiperazine **1** (with a load of 2 nmol mg^−1^). As demonstrated by Figure 7, the beads coated with compounds **3**–**5** gave similar results as the positive control.

## 3. Discussion

The data from the PAM fluorometry measurements (Figure 4) showed that diketopiperazine **7** caused a significant 10% reduction of Y_II_. Although it is unclear how compound **7** could affect the physiology of *S. robusta*, the results from the interference assay might be impaired due to the slightly toxic nature of **7**. The other analogs showed a small but non-significant inhibition of Y_II_. We therefore assumed that the use of these compounds would not alter the physiological behavior of *S. robusta*.

The impact of adding a diazirine (**3**) or an azide (**4** and **5**) to the cyclo(l-Pro-l-Pro) scaffold was rather limited. In case of the diazirine **3**, the interference at 100 nM equals that of **1** (100 nM). This suggests that the small, though polar, symmetrical increase in molecular volume of 3% (or 5.9 Å^3^) at position C-4 does not affect its activity (see Appendix A for molecular volumes and molecule models [59]). However, when the volume is further incremented, the activity of the compound decreases as demonstrated by acetal **8**. The two methoxy substituents increase the molecular volume by 52.7 Å^3^, yielding an overall volume increase of 30% compared to **1**.

In the case of the azides **4** and **5**, a slight difference between both diastereomers was observed. Our interference data suggests that a substitution of the azide on the pseudo-axial position of the l-proline ring (compound **4**), causes less steric hindrance between the ligand and its putative receptor than on the pseudo-equatorial position (compound **5**). The significance of this difference could not be determined because the compounds were evaluated in two independent experiments. Moreover, an interaction between the treatment and experiment is to be expected due to differences in diatom cell size and density.

The limited impact of the diazirine and azide moieties is also reflected by the outcome of the attraction assay. Beads coated with compounds **3**, **4,** or **5** had an attractiveness that was comparable to diketopiperazine **1**. The results of both assays demonstrate that these compounds could interact with the putative GPCR in the same manner as **1**.

The interference caused by hydroxy analog **6** was less pronounced. Lembke et al. reported that a symmetrical hydroxy analog did not show any activity [15]. Here we observed a significant interference, albeit only at the highest concentration tested. The introduction of a polar hydroxy group clearly decreases the pheromonal activity, but the pseudo-equatorial positioning of the substituent could be a contributing factor. In comparison: the pseudo-axial positioning of the methoxy group (**7**) seems to have less impact on the pheromonal activity. Yet, it is not clear how much of the interference effect of **7** is caused by the toxic nature of this compound.

The remaining azetidine analog **9**, with a two-carbon side-chain, showed a reduced activity compared to **1**, though still significant at 10 µM. This result validates the structural restrictions defined by Lembke et al. and suggests a preference of a three-carbon side-chain over a two-carbon side-chain [15].

While Figure 6 shows that the negative impact of some of the substitutions can be quite pronounced, all analogs (**3**–**9**) cause significant interference at 10 µM, which suggests that the modifications can still be tolerated. We also tried to evaluate whether the substitutions on the flanking ring structures impacted the conformation of the analogs’ central diketopiperazine ring, possibly affecting the interaction with the receptor.

Siemion and coworkers found an empirical relationship between the torsion of the C_β_-C_α_-C’=O fragment (θ) of the proline moiety and the difference between the ^13^C chemical shifts of C_β_ and C_γ_ in proline-containing cyclic peptides (Δδ_βγ_ = 0.081 |θ| + 2.47) [60]. The authors were also able to use the established relationship to determine the equilibrium between the boat and planar conformation of several proline-containing diketopiperazines [61].

The calculated torsion values of the pheromone analogs are smaller or equal to 30° and within the Δδ_βγ_ prediction interval for the postulated torsion angle of diketopiperazine **1** (4.91 ppm ± 1.48, see Appendix A) [60]. Based on these calculations, there are no indications that the adopted conformation of the central diketopiperazine ring of the diketopiperazines deviates from the preferred boat conformation of **1**. Additionally, there was no significant correlation found between the torsion angles and the bioactivity (data presented in Figure 6) of the analogs (Pearson correlation coefficient = 0.41, *p* = 0.3). These observations suggest that the introduced chemical moieties could have a minor to no effect on the diketopiperazine conformation.

Aside from the molecular dimensions and conformation, the substitutions also impact the lipophilic character of the pheromone analogs. The LogP values of the analogs were calculated (see Appendix A) and a positive correlation between these LogP values and the bioactivity (data presented in Figure 6) was found, however the correlation was not significant (Pearson correlation coefficient = 0.57, *p* = 0.14).

In conclusion, we have shown that it is possible to modify the pheromonal scaffold in an asymmetrical fashion, while retaining its pheromonal activity. Although structural restrictions demand hydrophobic ring flanking structures, nitrogen-containing substitutions, such as diazirines or azides, are tolerated as presented in the structure activity relationship map (Figure 8). The substitution with a hydroxy group seems to be less favored. The available space in the binding pocket of the putative pheromone receptor is limited, as demonstrated earlier by Lembke et al. [15], where a symmetrical variant of diketopiperazine **6** was found to be inactive. Though this could be a combined effect of the increased molecular volume and the less favored hydroxy substitution, the volumetric restriction reflects in our findings; the introduction of the acetal in diketopiperazine **8** largely reduces the bioactivity of the pheromone scaffold. The results of diazirine and azide analogs **3** to **5** demonstrate that these substituents can be incorporated into the cyclo(l-Pro-l-Pro) scaffold to probe the putative diatom pheromone receptor and to possibly characterize the action of **1** on other organisms as well.

## 4. Methods

### 4.1. Strain and Culture Conditions

*Seminavis robusta* DCG 0105 (85A, MT^+^) and DCG 0107 (85B, MT^−^) were used in this study. Both are available in the diatom culture collection of the Belgian Coordinated Collection of Microorganisms (BCCM/DCG, bccm.belspo.be/about-us/bccm-dcg) at Ghent University. Cultures were grown in artificial sea water: 34.5 g L^−1^ of Tropic Marin (Wartenberg, Germany), 0.08 g L^−1^ NaHCO_3_ (Sigma-Aldrich, St. Louis, Missouri, USA) enriched with Guillard’s F/2 solution (Sigma-Aldrich, St. Louis, Missouri, USA) at 20 mL L^−1^ inside a 250 mL cell culture flask. Light was provided in a 12 h:12 h light/dark cycle, using cool white fluorescent lamps (Philips, Amsterdam, Netherlands) with an intensity of 20 to 30 µmol m^−2^ s^−1^ and the temperature was held at 18 °C. Cultures were kept axenically by treating them regularly with antibiotics (0.5 g L^−1^ Penicillin, 0.5 g L^−1^ Ampicillin, 0.1 g L^−1^ Streptomycin and 0.05 g L^−1^ Gentamycin, Sigma-Aldrich, St. Louis, Missouri, USA). Stock cultures were kept at 4 °C with a light intensity of 2 to 5 µmol m^−2^ s^−1^. All cultures used in this study had a mean cell length between 20 and 35 µm.

### 4.2. Interference Assay

The interference assay was adopted from Lembke et al. [15] and a diagram of the assay is provided in the Appendix A. Oasis HLB beads were coated with cyclo(l-Pro-l-Pro) as described by Gillard et al., with a load of 2 nmol mg^−1^ [10]. In short, a 10 mg SPE cartridge was washed with methanol (1 mL) and Milli-Q water (2 mL). A 2 mL solution of cyclo(l-Pro-l-Pro) (20 nM) was eluted over the cartridge. The SPE material was suspended in 2 mL Milli-Q water and was stored as a stock solution at −20 °C in 50 µL aliquots. Before usage, a dilution was made by adding 20 µL of the stock suspension to 980 µL of Milli-Q water.

Axenic *S. robusta* 85A cultures were inoculated in a 24-well culture plate with a total of 1 mL medium per well and dark-synchronized for 40 h by wrapping the plate in aluminum foil. After synchronization, the cultures were conditioned by adding 1 mL 85B filtered medium (see Appendix A). Cyclo(l-Pro-l-Pro)–coated beads (50 µL of the diluted suspension) were added after 6 to 7 h and images were taken of the cell cultures with the BioTek Cytation3 Cell Imaging Multi-Mode Reader. For this study, pheromone (analogs) were applied in concentrations of 10 nM, 100 nM and 10 µM, starting from a 5 mM stock solution. After addition of the pheromone (analogs), the cell medium was mixed gently by pipetting the cell medium up and down.

In every interference experiment a positive and negative control is included. The positive control is the response of conditioned 85A cells towards cyclo(l-Pro-l-Pro)–coated beads, whereas the negative control assesses the attractiveness of non-coated beads.

Beads were assigned attractive or not attractive depending on a defined threshold, being a number of cells attached to a bead. In an effort to maximize the response difference between the positive control and the highest cyclo(l-Pro-l-Pro) **1** concentration tested (10 µM), an interference assay was set-up with 4 different cell densities. Aliquots of an exponentially growing stock culture were inoculated in a 24-well plate containing 2 mL of f/2 medium and incubated for 48 h. The cell medium was renewed with 1 mL of f/2 medium, cell densities were determined and the cultures were dark-adapted for 40 h. The interference test was carried out and the response towards the beads was assessed. The number of cells attached to every bead were evaluated after manual image capturing and categorized with the number attached cells varying from 1 to 9. The differences between both treatments, the positive control and 10 µM of cyclo(l-Pro-l-Pro) **1**, for every density and threshold were calculated (see Appendix A).

The highest difference in response can be found at a density of 7 × 10^7^ cells m^−2^ and a threshold of 1 cell per bead. The choice of a low threshold gives a higher difference between both treatments, but the downside is that the variance on the data is also higher (see Appendix A). This could explain the high variability in the interference test that was discussed in this study (Figure 5).

### 4.3. Attraction Assay

The attraction assay was adopted from Gillard et al. [10]. The procedure is similar to the interference assay described above, without the addition of (a) pheromone (analog) into the culture medium. Oasis HLB beads loaded with synthetic pheromone analogues **3**–**5** were made using the same procedure as for cyclo(l-Pro-l-Pro)–coated beads. Binding efficiencies were not determined.

### 4.4. Microscopy and Image Processing

#### 4.4.1. Automatic Image Capturing

Images were captured using a Cell Imaging Multi-Mode Reader (Cytation3, BioTek, Winooski, Vermont, USA), mounted with a 10x objective and controlled using Gen5 software (3.03.14, BioTek, Winooski, Vermont, USA). For the interference assays, image montages (7 by 7 images) were captured using three image channels: bright field, Cyanine 5 (CY5), and Green Fluorescent Protein (GFP). The CY5-channel was used to image the cell chlorophyll while the GFP-channel was used to image the beads. The bright field was used to autofocus the image and as reference. An algorithm was written in Python 3.6 (Python Software Foundation, https://www.python.org/) to assess the attractiveness of the beads (visualized in Figure 9). In short: both CY5 and GFP images were converted to a binary image after applying a threshold (Otsu’s method [62]). Next, beads were identified in the GFP image and if there was overlap between a bead and a cell, the bead was assigned as attractive. This method was used by default, unless mentioned otherwise.

#### 4.4.2. Manual Image Capturing

Images were captured on a Zeiss Observer A1 inverted microscope (Zeiss, Jena, Germany) equipped with a Zeiss EC plan-NEOFLUAR 10x or 4x lens (Zeiss, Jena, Germany) and coupled to a Nikon DS-Fi2 U3 camera (Nikon, Tokyo, Japan), operated by the NIS-elements software package (Nikon, Tokyo, Japan). Beads were classified attractive after manual evaluation.

### 4.5. Data Analysis

The statistical analysis of the interference assays was done in R 3.5 [63]. The number of attractive beads (summarized per well) were modeled as a Poisson generalized linear model in function of treatment (positive control, negative control and different compound effects) and a fixed plate effect that captures the variability between plates (replicates). The different treatment concentrations were modeled as separate treatment effects, effectively allowing for an interaction between the treatment and concentration. The total number of beads were added as offset to the model, thereby effectively modeling the fraction of attractive beads. In every experiment, one replicate per treatment was present per 24-well plate in a randomized fashion; four to six plates were used per experiment. Statistical inference was based on Wald tests using the asymptotic Chi-squared null distribution.

### 4.6. Phytotoxic Evaluation

The short-term phytotoxicity of the synthesized pheromone analogs was estimated using Pulse-Amplitude-Modulation fluorometry. Axenic *S. robusta* 85A cultures were inoculated in a 48-well plate (Greiner Bio-One, Kremsmünster, Austria). Diketopiperazine **1** and the seven analogs were added to a concentration of 10 µM. A treatment with PSII blocker DCMU (1 µM) was added as a control [57,58]. After addition of the compounds, the plate was incubated for 10 min at room temperature and Y_II_ was monitored with a MAXI Imaging PAM M-series fluorometer (Walz Mess- und Regeltechnik, Effeltrich, Germany) for 30 min, with a saturation pulse every 2 min and actinic light turned on (15 µmol photons m^−2^ s^−1^); sequential data were averaged over five replicates. The results are reported as relative inhibition of Y_II_, after interpolating the Y_II_ values of the pheromone analogs between those of **1** and DCMU (DCMU is 100% inhibition). The statistical analysis was done in R 3.5.2 using the lm() function.

## Data Availability

The data presented in this study are available in Appendix A.

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
