# Peer review of "Altering the Sex Pheromone Cyclo(l-Pro-l-Pro) of the Diatom *Seminavis robusta* towards a Chemical Probe"

_ijms, 2021, doi:10.3390/ijms22031037_

Round 1

Reviewer 1 Report

Modest Major Revisions:

While the figures and supporting information were prepared elegantly, the manuscript text fails to provide a proper discussion of the logic behind this program. It has been used to discuss what was done but little effort was provided to clearly explain the plan and integrate logical connections between the ideas. A small amount of effort should be able to rapidly tackle this problem and provide the readers with a superior manuscript.

I do think a small amount of effort could also be spent to improve the introduction as it currently reads like a ‘shopping list’ of papers to read instead of a scholarly presentation of the field.

It seems as if some picture or image of the bead assay would help. While one can read the Lembke article, the lack of at least some type of image within this manuscript makes challenges the reader to understand what is actually being done to screen these materials. For most readers, an additional figure within the text showing what these interference and attraction assays look like under a microscope would really help.

I must say I enjoyed the science completely, and one who understands the diversity and complexity of diketopiperazine chemistry and biology, I truly enjoyed this manuscript.

Minor Revisions:

Title: “Tweaking” is an informal word and should be replaced by a more appropriate medicinal chemical term.

Line 46 and throughout the manuscript: Terms such as i.e. and e.g. should be removed and the authors and the associated sentences should be rewritten to provide a concise meaning for what is provided as an example.

Line 53, the authors often refers to compound in a vague fashion. Instead of writing the diketopiperazine within this line and throughout this paragraph the authors should write 1 (in bold) or diketopiperazine 1 (in bold). This goes for all compound citations in the manuscript. Furthermore, the compound full name does not need to be repeated as it did in this paragraph.

Line 58, this sentence is very confusing; please restate, “The class of diketopiperazine compounds is represented by a plethora of metabolites within the biosphere”

Line 136, The following introduction is very weak, please expand on ‘The diazirine (3) was introduced as a photo-cross-linker and the azide (4,5) as a useful handle that is compatible with click chemistry.’ When was it done, by whom, and why is it so useful. While done at line 140 this paragraph and the one above it needs to be retooled so the reader understands where this is coming from.

Figure 3 should be provided with clear statistical criteria within the image as well as the caption. As shown, the images of each plot do not appear to be significant and the large expansion of the y-axis would raise concern from many readers

Figure 4. Draw out the entire structure at least once so one can see the SAR being evaluated. Ideally do so for compound 1.

Figure 4 and manuscript: the use of numbers for the experiments collides with the compound numbers adding unwanted confusion to the reader. Use either E1-E3 or A-C for the experiment numbering so it is easier to read. Also it would help if the text had a better description as to why multiple experiments are being presented as such.

Figures 5 7 need error bars, or a statement as to why they are not used

The authors use light orange in many figures for their bars and it is hard to see and does not print well. Please replace with a darker color.

The authors should create a final figure that shows their SAR map format PMID: 32273073 for excellent examples as to how to do this.

Gramatically, the authors often use the exact same sentence structure over and over for example, the term to characterize appears twice in lines 78-80. Again a small amount of editing could be conducted.

Reviewer 2 Report

This paper describes synthesis and bioevaluation of the potential chemical probes that can be used for biochemical studies of the diatom pheromone, diketopiperazine.

Though, it is difficult task to make usable chemical probes derived from small molecules such as proline diketopiperazine without losing its function. It is significant that the azide and diazirine derivatives, which have the most suitable functional groups for chemical probes that can be used in photoaffinity labeling and click chemistry, were found to exhibit the comparable activity with the original form.

Phototoxicity of the synthetic molecules have also been tested and confirmed to be non-toxix suggesting that these compounds are actually usable as chemical probes. Additional analyses of other derivatives as well as physicochemical properties including LogP values and steric hinderance also provided further information on the convertible chemical space for future derivative development.

Synthetic scheme is simple, easy, reliable and short, and all compounds were well identified based on the spectroscopic analyses. The bioassay is also well conducted with adequate control. Although, the bioassay results fluctuated due to using individual organisms, but it is acknowledged that sufficient activity is maintained.

Actual application of the synthetic molecules as chemical probes have not performed yet, but preparation of the promising probes is scientifically important step. Overall, the quantity and quality of the provided data is considered to be up to the standards, however, there is still room for improvement in the presentation. Generally, description of the manuscript is complicated and difficult to follow. Also, the method of the bioassay is difficult to image. There is no picture in spite of the bioassay was conducted based on the image analysis. After revision of the following points, I recommend to accept the manuscript.

The text is redundant for the content. It should be more concise. In the case of this paper, it might be better to put the results and discussion in the separate paragraphs.

Authors should add some cartoon showing the function of diketopiperazie in the diatom life cycle.

A diagram showing the bioassay method should be provided (supporting information is also acceptable).

A typical photo of the bioassay results should be shown.

Figure 7 should be placed in the Supporting Information.
